# Cytoprotective Effects of Delphinidin for Human Chondrocytes against Oxidative Stress through Activation of Autophagy

**DOI:** 10.3390/antiox9010083

**Published:** 2020-01-19

**Authors:** Dong-Yeong Lee, Young-Jin Park, Myung-Geun Song, Deok Ryong Kim, Sahib Zada, Dong-Hee Kim

**Affiliations:** 1Department of Orthopaedic Surgery, Armed Forces Daegu Hospital, Gyeongsan 38427, Korea; whatttary@hanmail.net; 2Department of Medicine, Gyeongsang National University School of Medicine, Jinju 52727, Korea; 3Department of Orthopaedic Surgery, Institute of Health Science, Gyeongsang National University Hospital and Gyeongsang National University School of Medicine, Jinju 52727, Korea; milesloner@gmail.com (Y.-J.P.); piano10000@naver.com (M.-G.S.); 4Department of Biochemistry and Convergence Medical Science, Institute of Health Science, Gyeongsang National University School of Medicine, Jinju 52727, Korea; drkim@gnu.ac.kr

**Keywords:** osteoarthritis, oxidative stress, apoptosis, autophagy, delphinidin, Nrf2

## Abstract

Antioxidant enzymes are decreased in osteoarthritis (OA) patients, implying the role of oxidative stress in osteoarthritis pathogenesis. The aim of this study was to evaluate the cytoprotective effects of delphinidin, a potent antioxidant, in human chondrocytes and the underlying mechanisms. The cytoprotective mechanism induced by delphinidin against oxidative stress (H_2_O_2_) in human chondrocytes was investigated. Cell viability and death were evaluated using proapoptotic and antiapoptotic markers such as cleaved caspase-3 (c-caspase-3), cleaved poly(ADP-ribose) polymerase *N*-acetylcysteine (c-PARP), Bcl-X_L_, and transcription factors associated with redox and inflammation regulation, including nuclear factor kappa B (NF-κB) and nuclear factor (erythroid-derived 2)-like 2 (Nrf2). Induction of autophagy was assessed by formation of LC3-II and autophagosome-(LC3 punctate, monodansylcadaverine (MDC) and acridine orange staining) in the presence or absence of an autophagy inhibitor. Treatment with delphinidin itself at concentration below 50 µM for 24 h did not affect viability of chondrocytes. Delphinidin inhibited reactive oxygen species (ROS)-induced apoptosis by significantly decreasing apoptosis markers such as c-caspase-3 and c-PARP while increasing antiapoptotic marker Bcl-X_L_ and antioxidant response NF-κB and Nrf2 pathways. Delphinidin also activated cytoprotective autophagy to protect chondrocytes during oxidative stresses. Activation of autophagy with autophagy inducer rapamycin also inhibited ROS-induced cell death and decreased proapoptotic proteins but increased antiapoptotic protein Bcl-X_L_, NF-κB, and Nrf2. Delphinidin can protect chondrocytes against H_2_O_2_-induced apoptosis via activation of Nrf2 and NF-κB and protective autophagy. Thus, it can inhibit OA with protection of chondrocytes. Delphinidin can protect chondrocytes against H_2_O_2_-induced ROS with maintenance of homeostasis and redox. These results suggest that delphinidin could be used to protect chondrocytes against age-related oxidative stress and other oxidative stresses in the treatment of OA. Thus, delphinidin may play a critical role in preventing the development and progression of OA.

## 1. Introduction

Osteoarthritis (OA) is the most common form of arthritis. It is also the most common cause of disability in older adults, affecting approximately 15% of the population. As an age-related disease, it mostly affects joints, shoulders, hands, feet, and spine, with knee and hip joints being commonly affected. The disease is characterized by deterioration of cartilage in joints, resulting in bones rubbing together that causes stiffness, pain, and impaired movement. Its risk factors include occupational injury, trauma, obesity, bone density, gender, lack of exercise, and genetic predisposition [1,2,3,4]. As a main cause of disability, OA reduces the quality of life with socio-economic burdens due to limited social activities, mood, memory loss, suicide, economic cost, and decreased productivity. OA is not only a problem in developing countries, but also a problem in developed countries [5,6,7,8,9,10]. To overcome these burdens, effective protective strategies need to be established by targeting leading causes of OA and improving the functional status of chondrocytes.

Age-related oxidative stress refers to increased intracellular levels of reactive oxygen species (ROS) known to play a critical role in the development and progression of OA [11,12]. Oxidative stress can lead to OA due to cell death of chondrocytes, telomerase shorting, DNA damage, and mitochondrial dysfunction, which can stimulate signaling pathways to initiate biological processes [11,12,13,14]. ROS can stimulate nuclear factor kappa B (NF-κB) and nuclear factor (erythroid-derived 2)-like 2 (Nrf2) signaling pathways. Both NF-κB and Nrf2 are transcription factors implicated in gene expression of antioxidant enzymes [15,16]. However, a high concentration of ROS causes apoptosis in chondrocytes [11,12,13,14]. NF-κB and Nrf2 signaling pathways contribute to ROS reduction and provide protection to chondrocytes against oxidative stress [15,16,17,18,19,20]. Thus, an effective protective strategy needs to be established by targeting oxidative stress through NF-κB and Nrf2 signaling pathways to maintain redox balance and chondrocyte homeostasis. 

ROS are known to be cytotoxic to most species including human. However, cells can neutralize ROS through molecular mechanisms using antioxidants and antioxidant enzymes. These antioxidants and antioxidant enzymes can convert superoxide to H_2_O and O_2_, regulate the intracellular redox, and counteract ROS to protect chondrocytes from oxidative damage [11,12,13,14,21,22]. Chondrocytes express a variety of antioxidants and antioxidants enzymes to prevent harmful effects of ROS. However, due to age-related decline in the expression of antioxidant enzymes and increased intracellular levels of ROS, homeostasis maintenance is lost and articular cartilage is degenerated, resulting in chondrocyte apoptosis and increased vulnerability to ROS-induced cell death [21,22,23]. Therefore, there is a need to search for ways protect chondrocytes against oxidative stress and maintain homeostasis. Recently, natural compounds such as cyanidin, delphinidin, malvidin, and pelargonidin were used as therapeutics for OA and several diseases [24]. Due to their potential as antioxidants, anthocyanins can protect chondrocytes against oxidative stress via various pathways, thus suppressing the progression of OA [25,26,27]. Among various kinds of anthocyanins, delphinidin is a specific class of polyphenols that can protect chondrocytes and inhibit the progression of OA [28]. However, how delphinidin protects chondrocytes and inhibits the progression of OA remains elusive.

Autophagy is a metabolic process that can sequester protein aggregates, damaged cellular organelles, and pathogen autophagosomes to be fused with lysosomes in the cytoplasm for degradation. This phenomenon is intrinsically adopted by cells to support them in a metabolically defiant microenvironment, to escape toxicity, and to reduce apoptosis induced by ROS [29,30]. Autophagy has dual and context-dependent roles in progression, protection, and death promotion in OA pathogenesis as in cancer [31,32,33]. Chondrocytes use autophagy as a very efficient housekeeping program for homeostasis maintenance and protection against various stresses [34,35,36,37]. The loss or impairment of autophagy in articular cartilage under mechanical or inflammatory stress was linked to aging-related cell death and increased OA severity [38,39,40,41,42,43]. However, the relationship between autophagy and cell death in chondrocytes and its mechanism are not fully understood yet. Additional studies are needed to decipher the underlying cell signaling mechanisms of autophagy.

In this study, we evaluated the potential role of delphinidin in protecting human chondrocytes against hydrogen peroxide (H_2_O_2_)-induced oxidative stress. Furthermore, we investigated which mechanism might be associated with delphinidin-induced cell protection in human chondrocytes. Cell survival is generally associated with apoptosis or autophagy. However, we do not yet know which pathway delphinidin uses to prevent the progression of OA. Thus, the purpose of this study was to investigate whether delphinidin might be cytoprotective for human chondrocytes and to further elucidate mechanisms associated with the effect of delphinidin. We hypothesized that delphinidin could prevent the progression of OA via autophagy.

## 2. Material and Methods

### 2.1. Reagents

C28/I2 human chondrocyte cells (SCC043) were purchased from Merck (Darmstadt, Germany). Fetal bovine serum (FBS; 16000-044), Dulbecco’s modified Eagle’s medium (DMEM, 11995-065), Roswell, and Hank’s buffered salt solution (HBSS, 14025-092) were purchased from Gibco and Life Technologies (Pittsburgh, PA, USA). Hydrogen peroxide solution (H_2_O_2_) (#MKBV3835V), *N*-acetylcysteine (NAC), DCFDA (2′,7′-dichloroflourescin diacetate), acridine orange hydrochloride hydrate (#318337), and monodansylcadaverine (MDC) (#30432) were bought from Sigma (St. Louis, MO, USA). Delphinidin chloride (cat no. ALX-385-028-M010, Lot no. L29971) was obtained from Enzo life sciences (Farmingdale, NY, USA). Chloroquine (C6628) was purchased from Sigma-Aldrich (St. Louis, MO, USA). Rapamycin (R-5000) was purchased from LC Laboratories (Woburn, England). CCK-8 (Cell Counting Kit-8) was purchased from Dojindo (Tokyo, Japan). The DeadEnd™ Fluorometric terminal uridine nick-end labeling (TUNEL) system (Ref#G3250, Lot #0000007545) was obtained from Promega (Madison, WI, USA.). The β-actin antibody (A5441) was from Sigma-Aldrich (St. Louis, MO, USA). Primary antibodies against LC3A/B (12741), caspase-3 (#9662), cleaved caspase-3 (#9661), NF-κB p65 (#8242), phospho-NF-κB p65 Ser536 (#3033), and cleaved poly(ADP-ribose) polymerase *N*-acetylcysteine (PARP) (#9541) were bought from Cell Signaling Technology (Beverly, MA, USA). Diamond Antifade Mountant with 4′,6-diamidino-2-phenylindole (DAPI) (#p36966) was purchased from Invitrogen (Calsbad, CA, USA). Antibodies against MAP1LC3 (Microtubule-associated proteins 1A/1B light chain 3B)(SC-376404) and Nrf2 (SC-722) were purchased from Santa Cruz biotechnology (Dallas, TX, USA). Secondary antibodies against rabbit immunoglobulin G (IgG) (STAR208P) and mouse IgG (STAR117P) were purchased from Bio-Rad (Hercules, CA, USA). Secondary antibodies for immunocytochemistry (fluorescein isothiocyanate, FITC) were obtained from Santa Cruz (Dallas, TX, USA).

### 2.2. Cell Viability

C28/I2 human chondrocyte cells (SCC043) were obtained from Merck and cultured at 37 °C in a 5% CO_2_ humidified atmosphere in DMEM medium supplemented with 5% (*v*/*v*) FBS, 100 units/mL penicillin, and 100 µg/mL streptomycin. Cell viability of C28/I2 human chondrocytes was determined using the CCK-8 kit. Cells (~5 × 10^4^ cells/well) in 96-well plates were treated with hydrogen peroxide solution (H_2_O_2_) in combination with other compounds such as NAC, delphinidin chloride, chloroquine, or rapamycin. Each well containing 100 µL of media and CCK-8 reagent (10 µL) was added into each well, and cells were further incubated at 37 °C for 4 h in a 5% CO_2_ humidified atmosphere. Cell viability was determined by measuring absorbance at 485 nm using a microplate reader (Hidex 1 FN/Chameleon; Turku, Finland).

### 2.3. Determination of Intracellular ROS

C28/I2 human chondrocyte (~5 × 10^4^ cells/well) were seeded in 96-well plates in DMEM medium (200 µL) supplemented with 5% FBS for 24 h. After incubation, cells were then treated with 500 µM H_2_O_2_ in the presence or absence of 40 µM of delphinidin and 5 mM NAC, incubated at 37 °C for 2 h. and 4 h. Intracellular ROS levels were determined with a DCFDA cellular ROS detection assay kit (cat. no. ab113851; Abcam, Burlingame, CA, USA) After adding 30 µM DCFDA dissolved in DMSO/phosphate-buffered saline (PBS) to each well, plates were incubated at 37 °C for 30 min in the dark. Plates were then read with a GloMax^®^ detection system (Model #E 8032; Promega, Sunnyvale, CA, USA) at 485/535 nm.

### 2.4. TUNEL Assay

TUNEL (terminal uridine nick-end labeling) assays were performed to measure the apoptosis by using the Promega DeadEnd™ Fluorometric TUNEL assay system. Cells were cultured on coverslips for 24 h and then treated with hydrogen peroxide solution (H_2_O_2_) in the presence or absence of NAC (*N*-acetylcysteine), delphinidin chloride, chloroquine, and rapamycin. Cells were fixed with 4% (*w*/*v*) paraformaldehyde for 30 min and permeabilized with PBS containing 0.1% Triton X-100 for 20 min at room temperature. Cells were blocked with 5% horse serum in PBS for 1 h, and then TUNEL assays were performed to measure the apoptosis accordance with the manufacturer’s instruction of the Promega DeadEnd™ Fluorometric TUNEL assay system for 1 h. After washing with PBS, cells the glass coverslips were mounted onto glass slides using a mounting medium containing DAPI. Slides were analyzed with a florescence microscope (BX51-DSU; Olympus, Tokyo, Japan). More than 1500 nuclei were counted per field. The experiment was repeated three times.

### 2.5. Western Blot Analysis

C28/I2 human chondrocytes, after treatment with hydrogen peroxide solution (H_2_O_2_) in the presence or absence NAC (*N*-acetylcysteine), delphinidin chloride, chloroquine, and rapamycin, were collected and washed twice with ice-cold 1× PBS. Total proteins were extracted with a cell lysis buffer supplemented with a protease and phosphatase inhibitor cocktail (Halt^TM^ Protease), and the Pierce bicinchoninic acid (BCA) protein assay kit (Thermo Scientific, Waltham, MA, USA) method was used for determination of protein concentration. After concentration, equal amounts (30 µg) of total protein were separated by 12% or 10% SDS-PAGE. Target proteins were specifically detected by Western blotting using indicated antibodies. Proteins were visualized with enhanced chemiluminescence detection reagent (Thermo Scientific, Waltham, MA, USA). All protein levels were normalized to β-actin levels.

### 2.6. Immunofluorescence Staining

C28/I2 human chondrocytes were cultured on polylysine-coated coverslips for 24 h. After incubation, cells were treated with 500 µM H_2_O_2_ in the presence or absence of 40 µM delphinidin chloride for 2 h. After treatment, cells were washed twice with 1× PBS and then fixed with 4% (*w*/*v*) paraformaldehyde for 30 min. After incubation, cells were washed with 1× PBS and permeabilized with PBS containing 0.1% Triton X-100 for 20 min at room temperature. Cells were blocked with 5% horse serum in PBS for 1 h and then incubated with primary antibodies overnight at 4 °C. After washing with PBS, cells were incubated with FITC-conjugated secondary antibodies (1:50 in PBS) at room temperature for 90 min. Slides were washed twice with PBS for 5 min each, and images were captured with a confocal microscope (FV-1000; Olympus, Tokyo).

### 2.7. Acridine Orange Dye Staining

C28/I2 cells were cultured on coverslips for 24 h and then treated with 500 µM H_2_O_2_ in the presence or absence of delphinidin for 2 h. After treatment, cells were washed three times with PBS, fixed with 4% formaldehyde, and permeabilized with 100% methanol. For determination of autophagy, an acridine orange (AO) assay were performed according to the given instructions. Acridine orange (AO) dye was added to cells at a final concentration of 1 μg/mL and incubated at room temperature for 15 min, protected from direct light. Red (acidic vacuoles) and green (cytoplasm) fluorescent images were captured with a confocal microscope (FV-1000; Olympus, Tokyo) using color filters for TRITC (Tetramethylrhodamine) and FITC (Fluorescein isothiocyanate), respectively. Overlapped images were then processed and quantified using NIH Image J software.

### 2.8. Monodansylcadaverine (MDC) Staining for Autophagic Vacuoles

C28/I2 cells were seeded into six-well plates with sterile cover slips for 24 h and then treated with 500 µM H_2_O_2_ in the presence or absence of delphinidin chloride, rapamycin, or chloroquine for 2 h. Following treatment, cells were washed with 1× PBS three times. After washing, autophagic vacuoles were labeled with MDC by incubating cells grown on coverslips with 0.05 mM MDC in PBS at 37 °C for 15 min in the dark (protected from direct exposure to light). After incubation, cells were washed with PBS four times and immediately analyzed with an inverted fluorescent microscope (Leica).

### 2.9. Statistical Analyses

Each experiment was done at least three times. Representative data are reported as means ± standard deviation (± SD). Differences between two groups were assessed using a two-tailed Student’s *t*-test. One-way analysis of variance (ANOVA) was used to compare means of three groups or more followed by a Tukey’s multiple comparison tests. Values of *p* < 0.05 and *p* < 0.01 were considered significant. All statistical analyses were performed using SPSS 18.0 for Windows (SPSS, Chicago, IL, USA).

## 3. Results

### 3.1. Delphinidin Has Cytoprotective Effects in Chondrocytes under Oxidative Stress

Firstly, to test the cytotoxic effect of H_2_O_2_ on human chondrocytes, cells were treated with 250 µM or 500 µM H_2_O_2_ in absence or presence of 5 mM NAC for 2 h and 4 h. Cell viability was significantly decreased after treatment with H_2_O_2_ in a time- and dose-dependent manner. However, cells treated with H_2_O_2_ in the presence of NAC showed a significant increase in cell viability (Figure 1A,B). Microscopic analysis of cell morphology and damage also revealed that H_2_O_2_ treatment was cytotoxic to these cells, while NAC had a cytoprotective effect against H_2_O_2_ treated cells, as shown in Figure 1B. The cytotoxic effect of H_2_O_2_ in these cells was caused by the accumulation of ROS (oxidative stress) since NAC, a well-known antioxidant, significantly increased the viability of H_2_O_2_-treated cells. 

To further investigate the cytoprotective role of delphinidin, we treated cells with different concentrations (10–75 uM) of delphinidin to check its cytotoxic effects. Interestingly, we did not observe any cytotoxicity of delphinidin until a concentration of 50 µM in 2, 4, and 24 h, with a slight decrease at 75 µM (Figure 1C). We used 40 µM delphinidin in all experiments hereafter.

To test the cytoprotective role of delphinidin, C28/I2 chondrocytes were treated with 500 µM H_2_O_2_ in the presence or absence of 40 µM delphinidin for 2 h and 4 h. As expected, the viability of H_2_O_2_-treated cells in the presence of delphinidin was significantly increased compared to cells treated with H_2_O_2_ in the absence of delphinidin. Microscopic analyses of cell morphology also suggested the cytoprotective role of delphinidin (Figure 1D,E). Taken together, these results suggest that delphinidin has a cytoprotective role in C28/I2 chondrocytes during oxidative stress.

To further investigate whether the cytoprotective role of delphinidin in chondrocytes might be due to its antioxidant activity, cells were treated with 500 µM H_2_O_2_ in the absence or presence 5 mM NAC (a well-known antioxidant) and 40 µM delphinidin. Interestingly, the relative ROS level in cells treated with H_2_O_2_ in the presence of NAC and delphinidin was significantly decreased compared to ROS level in cells treated with only H_2_O_2_ (Figure 1F). This result clearly suggests that the cytoprotective effect of delphinidin in chondrocytes is due to its antioxidant activity.

### 3.2. Delphinidin Inhibits Apoptotic Cell Death in Chondrocytes during Oxidative Stress

To investigate the effect of delphinidin on apoptosis of chondrocytes induced by H_2_O_2_, Western blot analysis was performed for proteins involved in apoptosis and cell protection (antioxidant response). Results showed that expression levels of proapoptotic proteins (c-caspase-3 and c-PARP) were significantly increased after treatment with H_2_O_2_. However, in the presence of delphinidin, expression levels of these proteins were sufficiently decreased (Figure 2A–C). Furthermore, levels of antiapoptotic protein Bcl-X_L_ and antioxidant response proteins (Nrf2 and p-NF-κB) were significantly decreased after treatment with H_2_O_2_. However, in the presence of delphinidin, levels of these proteins were sufficiently increased (Figure 2A,D–F). These results clearly suggest that delphinidin has an antiapoptotic role in chondrocytes via activation of Nrf2 and NF-κB. To further investigate the antiapoptotic role of delphinidin in chondrocytes, we performed a TUNEL assay to determine apoptosis. Interestingly, the percentage of H_2_O_2_-induced apoptosis of cells treated in the presence of delphinidin was significantly decreased compared to that of cells treated with H_2_O_2_ only (Figure 2G,H). These results further clarify the antiapoptotic role of delphinidin in chondrocytes.

### 3.3. Autophagy Is Involved in the Protection of Delphinidin against H_2_O_2_-Induced Oxidative Stress

To further test the cytoprotective activity of delphinidin, we measured autophagy induced by treatment with H_2_O_2_ in the presence and absence of delphinidin for 4 h. Firstly, we assessed the conjugation of LC3 (unconjugated LC3, called LC3-I) with PE (phosphatidylethanolamine) lipids, called LC3-II. Interestingly, expression levels of well-known autophagy marker LC3-II were significantly increased in the group treated with delphinidin only and in groups treated with H_2_O_2_ in the absence or presence of delphinidin (Figure 3A,B). These data were further confirmed by LC3 punctate analysis, as well as MDC and acridine orange (AO) staining analyses, which are indicative for the formation of autophagosomes or autolysosomes upon induction of autophagy. Once LC3s are conjugated with PE, they aggregate together into an intracellular double-membrane structure, called autophagosomes, and subsequently into autolysosomes fused with lysosomes. As shown above, delphinidin itself significantly increased LC3 puncta and further promoted these punctate numbers when chondrocytes were incubated with 40 µM delphinidin and 500 µM H_2_O_2_ (Figure 3C,D). In addition, MDC or AO dye staining analyses showed a similar formation of autolysosomes with treatment of delphinidin and hydrogen peroxide (Figure 3E–H). These results indicate that autophagy induced by delphinidin could be associated with the cytoprotective mechanism in chondrocytes in oxidative stress conditions.

### 3.4. Cytoprotective Role of Delphinidin Is Modulated by Activation of Autophagy in Oxidative Stress

Autophagy is a well-known stress response and a cytoprotective mechanism adopted by cells under various stresses. Given in these mechanisms, we further checked the functional role of autophagy in the antiapoptotic effect of delphinidin in chondrocytes treated with H_2_O_2_ in the absence or presence of 5 mM NAC, 40 µM delphinidin, autophagy activator rapamycin (100 nM Rapa), or inhibitor chloroquine (20 µM CQ). Interestingly, the viability of cells treated with H_2_O_2_ in the presence of 5 mM NAC, 40 µM delphinidin, and 100 nM Rapa was significantly increased compared to that of cells treated with H_2_O_2_ only. However, when cells were treated with H_2_O_2_ and 20 µM CQ, despite the presence of 5 mM NAC and 40 µM delphinidin, the viability of cells was significantly decreased compared to that of cells untreated with chloroquine (Figure 4A). The same results were observed in microscopic morphology examination of cells (Figure 4B). These results suggest that autophagy has a significant role in cell protection.

We also investigated the effect of autophagy in cell protection using the TUNEL assay. As expected, the percentage of apoptosis of TUNEL-positive cells treated with H_2_O_2_ in the presence of NAC was significantly decreased. Furthermore, delphinidin and autophagy activator rapamycin also significantly decreased the percentage of apoptosis compared to treatment with H_2_O_2_ only. However, the percentage of apoptosis in cells treated with H_2_O_2_ in the presence of autophagy inhibitor chloroquine, regardless of NAC or delphinidin, was significantly increased compared to that of cells treated with H_2_O_2_ only. These results further suggest the cytoprotective role of autophagy (Figure 4C,D). To investigate the cytoprotective role of autophagy activation involved in the effect of delphinidin against oxidative stress, we performed Western blot analysis for autophagy marker LC3, apoptosis markers c-caspase-3 and c-PARP, antiapoptotic marker Bcl-X_L_, and antioxidant response proteins (Nrf2 and p-NF-κB) (Figure 5A–F). Interestingly, levels of apoptosis markers such as c-caspase-3 and c-PARP were significantly decreased after treatment with H_2_O_2_ in the presence of NAC, delphinidin, or rapamycin compared to those in the group treated with H_2_O_2_ only (Figure 5A–C). On the other hand, levels of antiapoptotic protein (Bcl-X_L_) and antioxidant response proteins (Nrf2 and p-NF-κB) were significantly increased after treatment with H_2_O_2_ in the presence of NAC, delphinidin, or rapamycin compared to those in the group treated with H_2_O_2_ only (Figure 5A,D–F). However, inhibition of autophagy with CQ significantly increased the apoptosis of these cells in the group treated with H_2_O_2_ only or in the presence of NAC and delphinidin (Figure 5A–C), but significantly decreased levels of antiapoptotic protein Bcl-X_L_ and antioxidant response proteins (Nrf2 and p-NF-κB) (Figure 5A,D–F). These results clearly demonstrate that delphinidin can activate autophagy, which has a significant role in protecting humane chondrocytes against oxidative stress.

## 4. Discussion

The most important finding of this study was that delphinidin could protect human chondrocytes against oxidative stress by regulating Nrf2 and NF-κB and activating autophagy. Hydrogen peroxide (H_2_O_2_) has a cytotoxic effect on human chondrocytes via ROS activation, which can be inhibited by delphinidin through its antioxidant potential to protect cells from apoptosis. This protective mechanism of delphinidin was due to activation of antioxidant response pathways such as Nrf2 and NF-κB pathways and protective autophagy in chondrocytes (Figure 6). We hypothesized that delphinidin might have protective effects on human chondrocytes and help prevent the progression of OA by inducing autophagy. Results of this study support our hypothesis, indicating that delphinidin-induced protective effects for cells via autophagy might provide theoretical evidence and novel insight for treating OA.

This study demonstrated the molecular mechanism involved in the protection for chondrocytes elicited by delphinidin against H_2_O_2_-induced oxidative stress. Responses of cells to cytokines and growth factors depend on the cell’s redox status, which is the result of a subtle equilibrium between ROS production and intracellular antioxidant level. Previous studies reported that OA progression is significantly associated with oxidative stress and ROS [44,45]. In fact, ROS are produced at low levels in articular chondrocytes normally. Under normal circumstances, physiological levels of ROS can reversibly modify biomacromolecules which play crucial roles in regulating cell function. Furthermore, they are integral actors of intracellular signaling mechanisms contributing to the maintenance of cartilage homeostasis because they can modulate chondrocyte apoptosis, gene expression, extracellular matrix (ECM) synthesis and breakdown, and cytokine production [46,47,48,49]. However, under oxidative stress, excess ROS can affect chondrocytes in multiple ways, including telomerase shorting, DNA damage, mitochondrial dysfunction, and stimulation of cell death signaling pathways [11,12,13,14]. In the same vein, ROS levels due to oxidative stress were found to be elevated in patients with OA. Over-produced ROS can act as second messengers to promote the expression of matrix metalloproteinases (MMPs), which can lead to cartilage degeneration [50,51,52]. Furthermore, ROS can directly degrade matrix components and induce cell death [53]. Based on this theoretical evidence, the viability of cartilage cells was observed to be decreased when oxidative stress was applied to the cartilage in the present study. Furthermore, when using NAC and delphinidin as powerful antioxidants, cell viability was increased, confirming that H_2_O_2_-induced oxidative stress could directly affect the redox status of cartilage cells and become harmful to cells. 

Most existing pharmacologic therapies for OA such as oral administration of non-steroidal anti-inflammatory drugs (NSAIDs) and glucosamine are limited to pain management rather than prevention and cure. Surgery is typically the last resort for treating knee OA [54]. Indeed, efficient licensed disease-modifying drugs are currently unavailable. To overcome these burdens, effective protective strategies need to be established by targeting leading causes of OA and improving the functional status of chondrocytes. Recently, natural compounds such as anthocyanins were used as therapeutics for several diseases [24]. Among various anthocyanins, delphinidin is a specific class of polyphenols with beneficial effects on OA progression via various pathways. Wongwichai et al. [26] reported that anthocyanins can inhibit IL (interleukin)-1β induced expression of MMPs in human articular chondrocytes via NF-κB and the extracellular signal-regulated kinase/mitogen-activated protein kinase (ERK/MAPK) pathway. Haseeb et al. [28] reported that delphinidin can inhibit IL-1β-induced production of a cartilage-degrading molecule prostaglandin E2 (PGE2) via inhibition of cyclooxygenase-2 (COX-2) expression. Their results also verified that delphinidin could inhibit IL-1 receptor-associated kinase-1 (IRAK1^Ser376^) phosphorylation by IL-1β-induced activation of NF-κB in human chondrocytes. Although biomarkers such as IL-1β, IL-6, tumor necrosis factor-α (TNF-α), PGE2, and COX-2 related to anti-inflammation were not analyzed in the present study, results of this study confirmed that the mechanism via which delphinidin protected chondrocytes from oxidative stress was through underlying cell-death mechanisms.

In general, cell death is most commonly associated with apoptosis, although it can also occur by means of another mechanism such as autophagy. Apoptosis is a highly regulated, active process of cell death involved in the development, homeostasis, and aging [55]. Dysregulation of apoptosis can lead to pathological states such as various degenerative diseases. For instance, with increased oxidative stress, disequilibrium between DNA damage and repair can occur, resulting in cumulative un-repaired DNA damage. This accumulation could lead to altered gene transcription and the formation of altered proteins or the induction of apoptosis [56,57,58,59]. To prevent the development of OA, preventing the degradation of chondrocytes is essential. Thus, it is important to regulate the induction of excessive apoptosis. Unlike apoptosis, autophagy is a homeostatic mechanism and an important process in all cells to remove damaged mitochondria and misfolded proteins known to cause production of ROS that can cause DNA damage and lead to genomic instability [60]. However, excess autophagy induction can trigger autophagy-related apoptosis [61]. Thus, accurate understanding of the role of apoptosis and autophagy is critical for OA treatment.

Chondrocytes are pre-mitotic cartilage cells that lack the proliferative activity in adult articular cartilage, suggesting that there must be some strategies to protect chondrocytes cells under various stresses to resist apoptosis [14]. In this study, we found that delphinidin treatment increased levels of antiapoptotic protein (Bcl-X_L_, Nrf2, and NF-κB) but decreased levels of proapoptotic proteins (caspase-3, c-PARP) in chondrocytes. Furthermore, apoptosis was much decreased after treatment with delphinidin as compared to that in the group treated with H_2_O_2_ only. These results confirmed our hypothesis that delphinidin could be a potential therapeutic drug to prevent oxidative stress-induced cell death and protect chondrocytes. 

Nrf2 and NF-κB signaling pathways contribute to ROS reduction and provide protection to chondrocytes under oxidative stress [15,16,17,18,20]. Thus, an effective protective strategy could be established by targeting oxidative stress and Nrf2 and NF-κB signaling pathways to maintain redox balance and chondrocyte homeostasis. In the present study, we found that delphinidin could activate Nrf2 and NF-κB signaling pathways, suggesting the importance of Nrf2 and NF-κB signaling pathways in the protection of chondrocytes by delphinidin. A knockout mouse study showed that NF-κB has anti-apoptotic activity in non-immune cells. Many subsequent cell or tissue experiments showed that one of the most important functions of NF-κB in non-immune cells is its anti-apoptotic activity [62,63]. Nrf2 inductions may also play a key role in OA pathophysiology. Abnormal ROS signaling in OA by stress is associated with increased Nrf2 activity, which plays a major chondroprotective role in the progression of OA. It is also associated with suppressed levels of IL-1β-induced MMP-1, MMP-3, MMP-13, PGE2, and nitric oxide (NO) production [64]. Similar reports showed that other compounds with antioxidant properties such as hyaluronic acid (HA), 7,8-dihydroxyflavone (7,8-DHF), resveratrol, licochalcone A, diallyl disulfide, pterostilbene, wogonin, protandim, and 6-gingerol can exert anti-inflammatory and chondroprotective properties in joint tissues through activation of Nrf2 and NF-κB pathways [17,18,65,66,67,68,69,70]. These lines of evidence demonstrate the importance of Nrf2 and NF-κB signaling pathways in maintaining redox balance and chondrocyte homeostasis during aging and in OA. However, the precise signaling pathways that elicit these effects remain incompletely understood.

Autophagy has a dual and context-dependent role in progression, protection, and death promotion in OA pathogenesis as in cancer [31,32,33]. Chondrocytes use autophagy as a very efficient housekeeping program for homeostasis maintenance and protection under various stresses [34,35,36,37]. The loss or impairment of autophagy in articular cartilage under mechanical or inflammatory stress was linked to aging-related cell death and increased OA severity [38,39,40,41,42,43]. All these findings show that the relationship between autophagy and cell death in chondrocytes is not fully understood yet. Additional studies are needed to decipher the underlying cell signaling mechanisms of autophagy. Here, we found that cytoprotective mechanism of delphinidin was due to the activation of protective autophagy, since inhibition of autophagy increased apoptosis while its activation significantly decreased apoptosis. The finding that delphinidin has cytoprotective roles in oxidative stress of chondrocytes is valuable information. 

Although the present study has its merits, it also has several limitations. Firstly, it was found that redox signaling can control chondrocyte function by regulating signal transduction pathways such as those involving MAPK, phosphatidylinositol 3-kinase (PI3K)/protein kinase B (Akt), mammalian target of rapamycin (mTOR), and cyclic guanosine monophosphate (cGMP), as well as transcription factors such as hypoxia-inducible factor (HIF)-1a, activator protein-1 (AP-1), and NK-κB/p65 [71]. Although we used delphinidin as an antioxidant for redox control, due to limitations in using various antibodies, detailed aforementioned pathways were not proven. These detailed pathways need to be identified through further studies. Secondly, based on a previous study, OA cartilage is more sensitive to oxidative stress with significantly more ROS-induced DNA damage than normal cartilage [72]. However, this study only evaluated when delphinidin acted on oxidative stress in normal cartilage cells. Thirdly, for clinical application of delphinidin as a chemotherapeutic agent, although the present studies documented the cytoprotective effects on chondrocytes in vitro by delphinidin, concentrations of bioactive compounds sufficient enough to exert such protective effects in vivo need to be determined. For instance, the bioeffective compounds might be formed or removed in vivo due to intestinal bacterial and/or hepatic metabolism [73]. For these reasons, to obtain the concentrations required to induce direct cell-protective activity is difficult in vivo. This is thought to be closely related to the method of application (topical, oral, or injection) of bioactive compounds and should be proven through animal models and pre-clinical trials on the basis of our findings. Our long-term clinical goal is to reduce the prevalence of osteoarthritis by demonstrating how delphinidin can prevent OA progression. Therefore, the precise mechanism and application via which delphinidin operates in OA cartilage also need to be determined in the future. Despite these limitations, this study is the first to show that delphinidin can protect cells against oxidative stress by inhibiting apoptosis and causing autophagy.

## 5. Conclusions

Delphinidin can protect chondrocytes from H_2_O_2_-induced apoptosis by activating Nrf2 and NF-κB and inducing protective autophagy in chondrocytes, thus inhibiting OA. These results suggest that delphinidin could be used to protect chondrocytes from age-related oxidative stress and other oxidative stresses for treating OA. Thus, our results indicate that delphinidin may play a critical role in preventing the development and progression of OA.

## Figures and Tables

**Figure 1 antioxidants-09-00083-f001:**
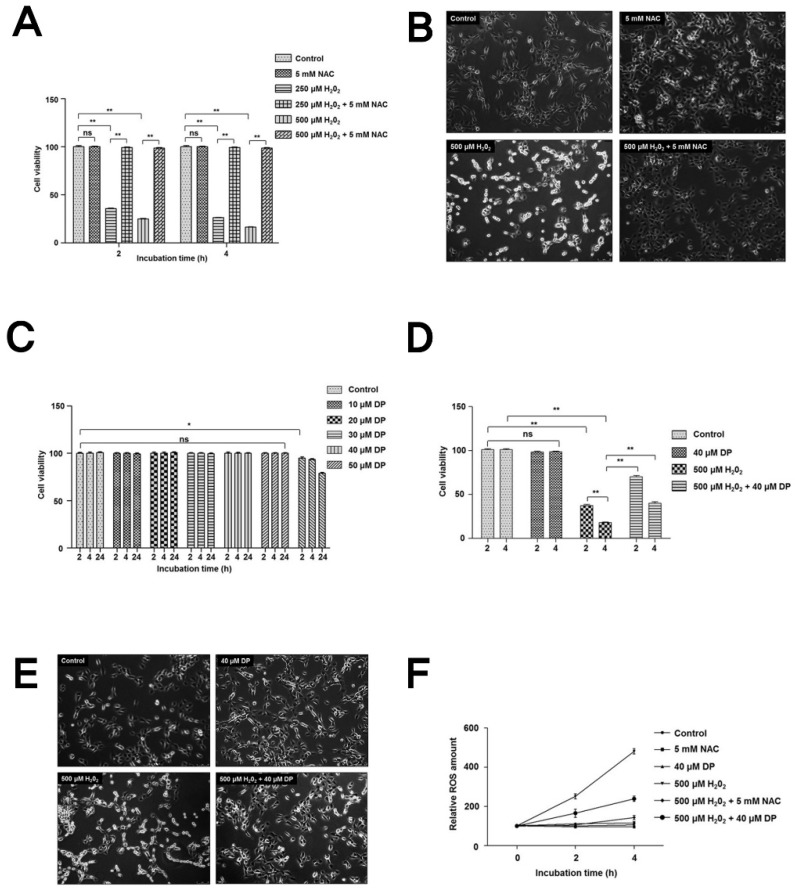
Delphinidin protects C28/I2 chondrocyte cells in hydrogen peroxide cytotoxicity. (**A**) Determination of cell viability. The C28/I2 chondrocyte cells were treated with 250 µM and 500 µM H_2_O_2_ in presence or absence of 5 mM *N*-acetylcysteine (NAC) for 2 h and 4 h. Cell viability was determined by the Cell Counting Kit-8 (CCK-8) assy. Data represent the means (± SD) of three independent experiments (** *p* < 0.01; ns indicates not significant). (**B**) C28/I2 chondrocyte cells treated with 500 µM H_2_O_2_ for 4 h; cells images were analyzed for cell morphology (100× magnification) using bright-field microscopy at 2 h (scale bar = 10 µm). (**C**) Titration of delphinidin for cytotoxicity in C28/I2 cells. C28/I2 cells were treated with different concentration of delphinidin (DP, 10–75 µM) for 2 h, 4 h, and 24 h. Cell viability was determined by the CCK-8 assay. Data represent the means (± SD) of three independent experiments (* *p* < 0.05; ns indicates not significant). (**D**) The C28/I2 chondrocyte cells were treated with 500 µM H_2_O_2_ in the presence or absence of 40 µM delphinidin for 2 h and 4 h. Cell viability was determined by the CCK-8 assay. Data represent the means (± SD) of three independent experiments (** *p* < 0.01; ns indicates not significant). (**E**) Treated cell images were analyzed for cell morphology (100× magnification) using bright-field microscopy at 4 h (scale bar = 10 µm). (**F**) C28/I2 cells were incubated with 500 µM H_2_O_2_ in the presence or absence of NAC and delphinidin for the indicated time periods. The relative intracellular reactive oxygen species (ROS) at each time point were determined using a 2′,7′-dichloroflourescin diacetate (DCFDA) assay; statistical analysis was performed by one-way ANOVA; * *p* < 0.05 was considered significant.

**Figure 2 antioxidants-09-00083-f002:**
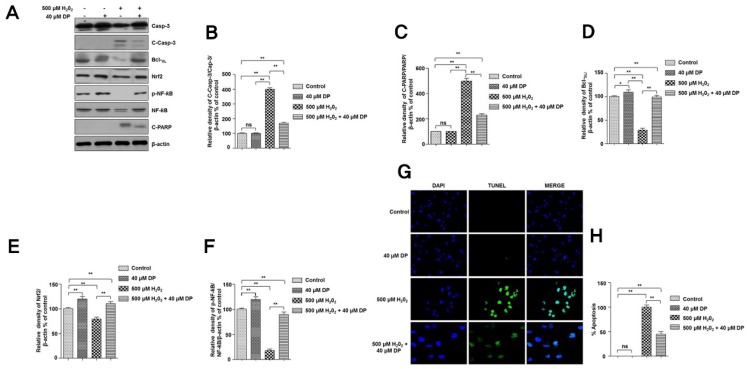
Delphinidin protects C28/I2 chondrocyte cells from H_2_O_2_-induced oxidative stress via nuclear factor (erythroid-derived 2)-like 2 (Nrf2) and nuclear factor kappa B (NF-κB). (**A**) The C28/I2 chondrocyte cells were treated with 500 µM H_2_O_2_ in presence or absence of 40 µM delphinidin (DP) for 4 h. After cell lysis, total cell extracts (30 µg) were separated on 8% or 10% SDS-PAGE and analyzed by Western blotting using primary antibodies against proteins (Bcl-X_L_, caspase-3, cleaved caspase-3, Nrf2, NF-κB, p-NF-κB, and cleaved poly(ADP-ribose) polymerase *N*-acetylcysteine (PARP)). β-Actin was used as a loading control. (**B**–**F**) The relative amounts of caspase-3/leaved caspase-3, cleaved PARP, Bcl-X_L_, Nrf2, NF-κB, and p-NF-κB, respectively, shown in Western blot analyses, were quantified by NIH ImageJ software and represented as a graph. Data represent the means (± SD) of three independent experiments (** *p* < 0.01, * *p* < 0.05; ns indicates not significant). (**G**–**H**) The C28/I2 chondrocyte cells were treated with 500 µM H_2_O_2_ in the presence or absence of 40 µM delphinidin (DP) for 4 h. A terminal uridine nick-end labeling (TUNEL) assay was preformed after according to the instructions of the Promega DeadEnd™ Fluorometric TUNEL system kit. The images were captured using a florescence microscope (BX51-DSU; Olympus, Tokyo). Data represent the means (± SD) of three independent experiments (** *p* < 0.01; ns indicates not significant).

**Figure 3 antioxidants-09-00083-f003:**
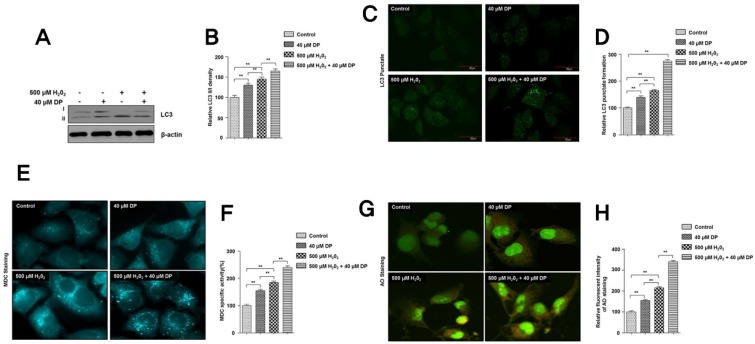
Delphinidin induces autophagy and protects C28/I2 chondrocyte cells from H_2_O_2_-induced oxidative stress cell death. (**A**,**B**) The C28/I2 chondrocyte cells were treated with 500 µM H_2_O_2_ in the presence or absence of 40 µM delphinidin (DP) for 4 h. After cell lysis, total cell extracts (30 µg) were separated on 8% or 10% SDS-PAGE and analyzed by Western blotting using primary antibodies against LC3 protein. The relative level of LC3-II to LC3-I was quantified (**B**). β-Actin was used as a loading control. (**C**,**D**) C28/I2 chondrocyte cells were cultured on coverslips for 24 h and then treated with 500 µM H_2_O_2_ in the presence or absence of 40 µM delphinidin (DP) for 4 h. After fixing cells with paraformaldehyde, cells were incubated for 24 h at 4 °C with mouse monoclonal anti-LC3 antibody. After washing, secondary antibodies (fluorescein isothiocyanate (FITC)-conjugated anti-mouse antibodies) were applied to cells. The glass slides were mounted using a mounting medium containing 4′,6-diamidino-2-phenylindole (DAPI) (to stain nuclei), and all images were captured by confocal microscopy (Olympus FV-1000). LC3 punctate was quantified using NIH ImageJ software. Data represent the means (± SD) of three independent experiments (** *p* < 0.01). (**E**,**F**) For the monodansylcadaverine (MDC) assay, chondrocyte cells were treated with H_2_O_2_ in the presence or absence of 40 µM delphinidin for 4 h. Following this incubation period, both cells were incubated with MDC at 0.05 mM for 10 min at 37 °C and then washed four times with phosphate-buffered saline (PBS) pH 7.4. Cells were immediately analyzed by fluorescence microscopy and quantified (**F**) as described in Section 2 for MDC-labeled autophagic vacuoles. (**G**,**H**) Acridine orange (AO) dye staining for autophagic vacuoles was carried out in chondrocytes cells which were treated with H_2_O_2_ in the presence or absence of 40 µM delphinidin (DP) for 4 h. Following this incubation period, both cells were incubated with AO for acidic vacuolar organelles. Cells were immediately analyzed by fluorescence microscopy and quantified (**H**) as described in Section 2 for AO-labeled autophagic vacuoles, ** *p* < 0.01.

**Figure 4 antioxidants-09-00083-f004:**
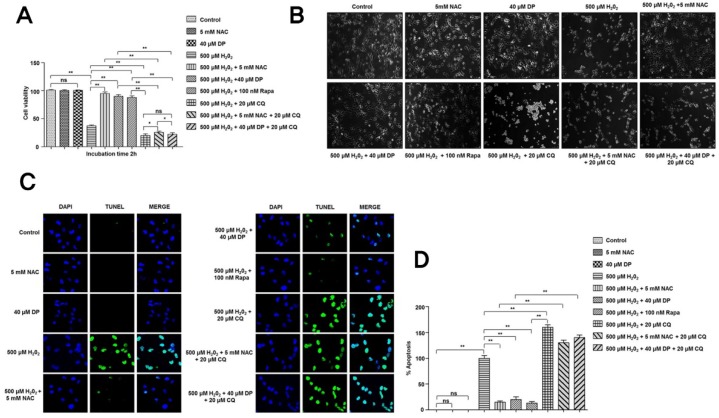
Inhibition of delphinidin-induced autophagy increases H_2_O_2_-induced apoptosis in C28/I2 chondrocyte cells. (**A**,**B**) The C28/I2 chondrocyte cells were treated with 500 µM H_2_O_2_ in the presence or absence of 40 µM delphinidin (DP), 100 nM rapamycin (Rapa), and 20 µM chloroquine (CQ) for 4 h. Cell viability was determined by the CCK-8 assay. Data represent the means (± SD) of three independent experiments (** *p* < 0.01, * *p* < 0.05; ns indicates not significant). Treated cell images were analyzed for cell morphology (100× magnification) using bright-field microscopy at 2 h (ns indicates not significant; scale bar = 10 µm). (**C**,**D**) The C28/I2 chondrocyte cells were treated with 500 µM H_2_O_2_ in the presence or absence of 40 µM delphinidin (DP), 100 nM rapamycin (Rapa), and 20 µM chloroquine (CQ) for 4 h. TUNEL assays were preformed after according to instructions of the Promega DeadEnd™ Fluorometric TUNEL system kit. The images were captured using a florescence microscope (BX51-DSU; Olympus, Tokyo). Data represent the means (± SD) of three independent experiments (** *p* < 0.01; ns indicates not significant).

**Figure 5 antioxidants-09-00083-f005:**
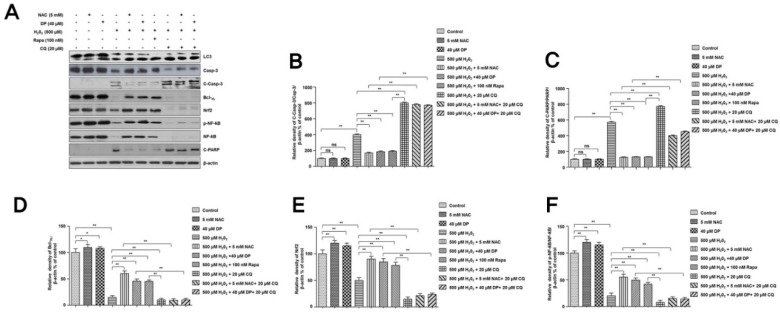
Inhibition of delphinidin-induced autophagy increases H_2_O_2_-induced apoptosis in C28/I2 chondrocyte cells. (**A**) The C28/I2 chondrocyte cells were treated with 500 µM H_2_O_2_ in the presence or absence of 40 µM delphinidin (DP), 100 nM rapamycin (Rapa), and 20 µM chloroquine (CQ) for 4 h. After cell lysis, total cell extracts (30 µg) were separated on 8% or 10% SDS-PAGE and analyzed by Western blotting using primary antibodies against proteins (LC3, Bcl-X_L_, caspase-3, cleaved caspase-3, Nrf2, NF-κB, p-NF-κB, and cleaved PARP). β-Actin was used as a loading control. (**B**–**F**) Quantifications of protein expression and activation. The relative amounts of all proteins (Bcl-X_L_, caspase-3, cleaved caspase-3, Nrf2, NF-κB, p-NF-κB, and cleaved PARP, respectively) shown in Western blot analyses were quantified by NIH ImageJ software and represented as a graph. Data represent the means (± SD) of three independent experiments (* *p* < 0.05, ** *p* < 0.01; ns indicates not significant).

**Figure 6 antioxidants-09-00083-f006:**
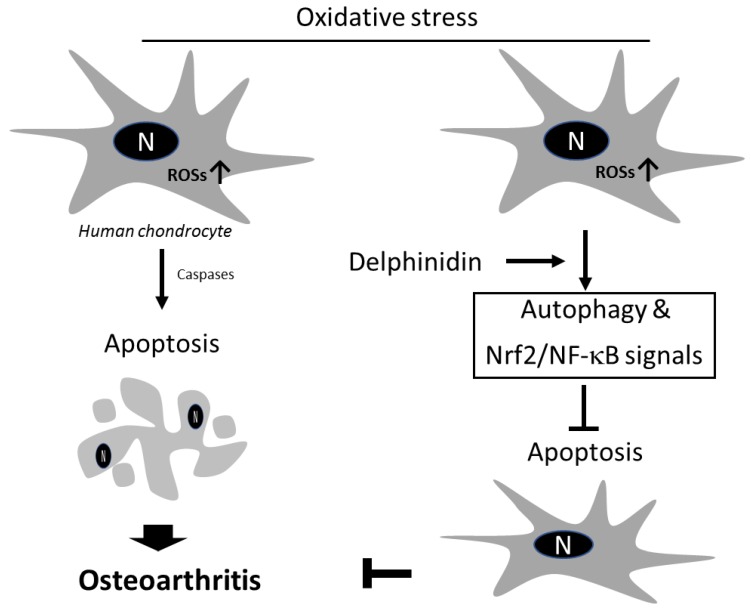
Schematic representation of delphinidin protective role against H_2_O_2_-induced apoptosis in C28/I2 chondrocyte cells. H_2_O_2_-induced apoptosis via accumulation of intracellular reactive oxygen species (ROS) in C28/I2 cells, leading to osteoarthritis pathogenesis. Delphinidin protects chondrocytes from ROS-induced apoptosis via activation of Nrf2 and NF-κB pathways and protective autophagy. Arrow-headed and bar-headed lines indicate activation and inhibition during the process, respectively.

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
