# Peer review of "Cytoprotective Effects of Delphinidin for Human Chondrocytes against Oxidative Stress through Activation of Autophagy"

_antioxidants, 2020, doi:10.3390/antiox9010083_

Round 1
Reviewer 1 Report
The authors tried to find the effects of delphinidine on chondrocyte viability and autophag formation. The process of experiment was appropriate which overall made for a very interesting read with strong results. I only have minor suggested corrections.
500 µM H2O2 is very high, is there any reference to prove its physiologycal effects?
In Figs 2A and 5A molecular weight should be written on the figure.
Minor suggestion:
in line 124 "Cells (~5x10... )" space is missing
in line 221 "with 500 µM" space is missing
in line 269 "confirmed GFP-LC3" space is missing
in line 274 "were incubated with 40 M delphinidin and 500 M H2O2" Are these concentrations correct?
in line 306 and 309 "40 M delphinidin " or "20 M CQ" Is this concentration correct?
Author Response
Reviewer 1
The authors tried to find the effects of delphinidin on chondrocyte viability and autophagy formation. The process of experiment was appropriate which overall made for a very interesting read with strong results. I only have minor suggested corrections.
500 µM H2O2 is very high, is there any reference to prove its physiological effects?
Answer) The reviewer is right and I appreciate his point of view that 500 µM H2O2 is a high concentration but it depends upon the incubation time of treatment. As like with 4h treatment almost 50 % cells were died however with treatment of 24h almost much cells were died (result not shown). That why we have selected 500 µM H2O2 for 4h.
In Figs 2A and 5A molecular weight should be written on the figure.
Answer) The reviewer question is good but most of lab published manuscripts and in other papers didn’t mention the molecular weight (Kd). As they make a crowed so we didn’t mentioned molecular weight (Kd), indeed we have mentioned about all protein and antibodies with their catalog numbers so any one can get any information.
Minor suggestion:
in line 124 "Cells (~5x10... )" space was missing
Answer) I agree. Space is added (in line 125).
in line 221 "with 500 µM" space is missing
Answer) I agree. Space is added (in line 228).
in line 269 "confirmed GFP-LC3" space is missing
Answer) I agree. Space is added (in line 277).
in line 274 "were incubated with 40 M delphinidin and 500 M H2O2" Are these concentrations correct?
Answer) I’m sorry that. “40 M delphinidin and 500 M H2O2” was changed to “40 µM delphinidin and 500 µM H2O2” (in line 283)
in line 306 and 309 "40 M delphinidin " or "20 M CQ" Is this concentration correct?
Answer) It is incorrect. "40 M delphinidin " and "20 M CQ" were changed to "40 µM delphinidin " or "20 µM CQ" (in line 315-319)

Reviewer 2 Report
I have read the paper entitled Cytoprotective effects of delphinidin for human 2 chondrocytes against oxidative stress through activation of autophagy by Dong-Yeong and collaborators. The paper assesses that Delphinidin can protect chondrocytes against H2O2-induced ROS with maintenance of homeostasis and redox.
The paper sounds good. this review has only an important concern:
Authors assess that delphinidin may play a critical role as a chemotherapeutic agent to prevent the development and progression of OA. This assertion is too strong. They did not even perform preclinical experiments, so this reviewer suggests to be less assertive and more probabilistic along the text.
Author Response
Reviewer 2
I have read the paper entitled “Cytoprotective effects of delphinidin for human chondrocytes against oxidative stress through activation of autophagy” by Dong-Yeong and collaborators. The paper assesses that Delphinidin can protect chondrocytes against H2O2-induced ROS with maintenance of homeostasis and redox.
The paper sounds good. This review has only an important concern:
Authors assess that delphinidin may play a critical role as a chemotherapeutic agent to prevent the development and progression of OA. This assertion is too strong. They did not even perform preclinical experiments, so this reviewer suggests to be less assertive and more probabilistic along the text.
Answer) I agree your opinion. We are in the process of animal study and clinical research in relation to this subject. The results of this study alone are exaggerated to claim “Thus, delphinidin may play a critical role as a chemotherapeutic agent to prevent the development and progression of OA.” Therefore, the sentence was changed to “Thus, delphinidin may play a critical role to prevent the development and progression of OA.”
